

# Regenerative agriculture: merging farming and natural resource conservation profitably

Claire E. LaCanne[1] and Jonathan G. Lundgren[2]

[1] Natural Resource Management Department, South Dakota State University, Brookings, SD, USA
[2] Ecdysis Foundation, Estelline, SD, USA

## ABSTRACT

Most cropland in the United States is characterized by large monocultures, whose productivity is maintained through a strong reliance on costly tillage, external fertilizers, and pesticides (*Schipanski et al., 2016*). Despite this, farmers have developed a regenerative model of farm production that promotes soil health and biodiversity, while producing nutrient-dense farm products profitably. Little work has focused on the relative costs and benefits of novel regenerative farming operations, which necessitates studying *in situ*, farmer-defined best management practices. Here, we evaluate the relative effects of regenerative and conventional corn production systems on pest management services, soil conservation, and farmer profitability and productivity throughout the Northern Plains of the United States. Regenerative farming systems provided greater ecosystem services and profitability for farmers than an input-intensive model of corn production. Pests were 10-fold more abundant in insecticide-treated corn fields than on insecticide-free regenerative farms, indicating that farmers who proactively design pest-resilient food systems outperform farmers that react to pests chemically. Regenerative fields had 29% lower grain production but 78% higher profits over traditional corn production systems. Profit was positively correlated with the particulate organic matter of the soil, not yield. These results provide the basis for dialogue on ecologically based farming systems that could be used to simultaneously produce food while conserving our natural resource base: two factors that are pitted against one another in simplified food production systems. To attain this requires a systems-level shift on the farm; simply applying individual regenerative practices within the current production model will not likely produce the documented results.

Corresponding author
Jonathan G. Lundgren,
jgl.entomology@gmail.com

## INTRODUCTION

Development of synthetic fertilizers, hybrid crops, genetically modified crops, and policies that decouple farmer decisions from market demands all helped create a modern food production system which reduces the diversity of foods that are produced (*Fausti & Lundgren, 2015*; *Pretty, 1995*). This simplification of our food system contributes to climate change (*Carlsson-Kanyama & Gonzalez, 2009*), rising pollution (*Beman et al., 2011*; *Morrissey et al., 2015*), biodiversity loss (*Butler, Vickery & Norris, 2007*; *Landis et al., 2008*),

and damaging land use changes (*Johnston, 2014*; *Wright & Wimberly, 2013*) that affect the sustainability, profitability and resilience of farms (*Schipanski et al., 2016*). Farmers experience the highest suicide rate of any profession in the United States, a rate nearly five-fold higher than the general public (*McIntosh et al., 2016*); the driving depression rates are related to conventional production practices (*Beard et al., 2014*). The scale of our food production system provides opportunities for solving some of these planetary scale problems (*Lal, 2004*; *Teague et al., 2016*), but requires a systems-level shift in the values and goals of our food production system that de-prioritizes solely generating high yields toward one that produces higher quality food while conserving our natural resource base.

The goal of regenerative farming systems (*Rodale, 1983*) is to increase soil quality and biodiversity in farmland while producing nourishing farm products profitably. Unifying principles consistent across regenerative farming systems include (1) abandoning tillage (or actively rebuilding soil communities following a tillage event), (2) eliminating spatio-temporal events of bare soil, (3) fostering plant diversity on the farm, and (4) integrating livestock and cropping operations on the land. Further characterization of a regenerative system is problematic because of the myriad combinations of farming practices that comprise a system targeting the regenerative goal. Other comparisons of conventional agriculture with alternative agriculture schemes do not compare *in situ* best management practices developed by farmers, and frequently ignore a key driver to decision making on farming operations: the examined systems' relative net profit to the farmer (*De Ponti, Rijk & Van Ittersum, 2012*).

## MATERIALS AND METHODS

Corn (*Zea mays* L.) was selected for our study due to its pre-eminence as a food crop in North America and globally. Corn is planted on 39.9% of all crop acres (*NASS, 2017*), or 4.8% (37.1 million ha) of the terrestrial land surface of the contiguous 48 states. In 2012, it generated 30.3% ($64,319 billion) of all gross crop value in the US (*NASS, 2017*). Nearly 100% of cornfields are treated annually with insecticides (*NASS, 2017*). We used a matrix of specific production practices (Table 1) to define each farm into one of two systems (regenerative or conventional). The most regenerative systems ($n = 40$ fields on 10 farms) used mixed multispecies cover crops (ranging from 2–40 plant species), were never-till, used no insecticides, and grazed livestock on their cropland. The most conventional farms practiced tillage at least annually (36 fields on eight farms), applied insecticides (as GM insect-resistant varieties and neonicotinoid seed treatments), and left their soil bare aside from the cash crop.

Soil organic matter, insect pest populations, and corn yield and profit were assessed for each field. Soil cores (8.5 cm deep, 5 cm in diameter; 30 g of soil each; $n = 4$ samples per field that were made a composite sample; only one field was sampled per farm-selected by the producer- and two farms were omitted due to adverse weather during the sampling event) were collected at least 10 m from one another during anthesis. Samples were cleaned of plant residue, ground, and dried to constant weight at 105 °C. Particulate soil organic matter (POM) was determined by screening each sample (soaked in 5 g L$^{-1}$

**Table 1  Trait matrix used to assign farms to regenerative or conventional corn production systems.** The composite rank scores are based on the number of regenerative practices used on a particular farm. Farms whose rank scores are in the top 50% of farms are considered regenerative (shaded rows); those with rank scores in the lower half are conventional (white rows). To aid interpretation, additional traits of each system could be included in enhanced trait matrices. Organic operations are indicated by an asterisk in the "Reference town" column.

| Reference town | Farm locations (latitude, longitude) | Cover crop (yes: 1; no: 0) | Insecticide (no: 1; yes: 0) | Other pesticides (no: 1; yes: 0) | Tillage (yes: 0; no: 1) | Grazed corn field (yes: 1; no: 0) | Composite rank score |
|---|---|---|---|---|---|---|---|
| Bladen, NE | 40.31971, −98.57358 | yes | no | yes | no | no | 3 |
| Bladen, NE | 40.33703, −98.56301 | no | yes | yes | yes | no | 0 |
| York, NE | 40.63054, −97.66534 | yes | no | yes | no | no | 3 |
| York, NE | 40.97390, −97.49031 | no | yes | yes | yes | no | 0 |
| Bismarck, ND | 46.85280, −100.60131 | yes | no | no | no | yes | 5 |
| Bismarck, ND | 46.85280, −100.35145 | no | yes | yes | no | no | 1 |
| Bismarck, ND | 46.81734, −100.51257 | yes | no | yes | no | yes | 4 |
| Bismarck, ND | 47.14250, −100.19720 | no | yes | yes | no | no | 1 |
| White, SD* | 44.42572, −96.58806 | yes | no | no | yes | no | 3 |
| White, SD | 44.41155, −96.60008 | no | yes | yes | yes | no | 0 |
| Pipestone, MN* | 44.11446, −96.32468 | yes | no | no | yes | no | 3 |
| Pipestone, MN | 44.12416, −96.36422 | no | yes | yes | yes | no | 0 |
| Toronto, SD | 44.59248, −96.57923 | yes | yes | yes | no | no | 3 |
| Toronto, SD | 44.57960, −96.58367 | no | yes | yes | yes | no | 0 |
| Gary, SD* | 44.80565, −96.34708 | yes | no | no | yes | yes | 4 |
| Gary, SD | 44.80689, −96.35465 | no | yes | yes | yes | no | 0 |
| Arlington, SD | 44.41566, −97.18795 | yes | no | yes | no | yes | 4 |
| Arlington, SD | 44.42644, −97.25077 | no | yes | yes | yes | no | 0 |
| Lake Norden, SD | 44.58976, −97.08649 | yes | yes | yes | no | yes | 3 |
| Lake Norden, SD | 44.55.6839, −97.243820 | no | yes | yes | yes | no | 0 |

aqueous hexametaphosphate) through 500 um (course POM) and 53 um (fine POM) sieves and then applying the loss on ignition (LOI) technique (*Davies, 1974*). Insect pests were enumerated through dissections of all aboveground plant tissues (25 plants per field). Major pests of corn (rootworm adults, caterpillar pests, and aphids) are all present in cornfields at this crop developmental stage (*Lundgren et al., 2015*), and this was substantiated in the observations in this study as well. Yields were gathered from three randomly selected 3.5 m sections of row from each field. Gross revenue for each field were considered as yield and return on grain, and additional revenue streams (e.g., animal weight gain resulting from grazing). Total direct costs for each field were calculated based on the costs of corn seed, cover crop seed, drying/cleaning grain, crop insurance, tillage, planting, fertilizers, pesticides, and irrigation.

## RESULTS AND DISCUSSION

Insect pest populations were more than 10 fold higher on the insecticide-treated farms than on the insecticide-free regenerative farms (ANOVA; $F_{1,77} = 13.52$, $P < 0.001$; Fig. 1). Pest populations were numerically dominated by aphids, but each of the individual pest species followed the same pattern of the aggregated data; none of these pests were at economically

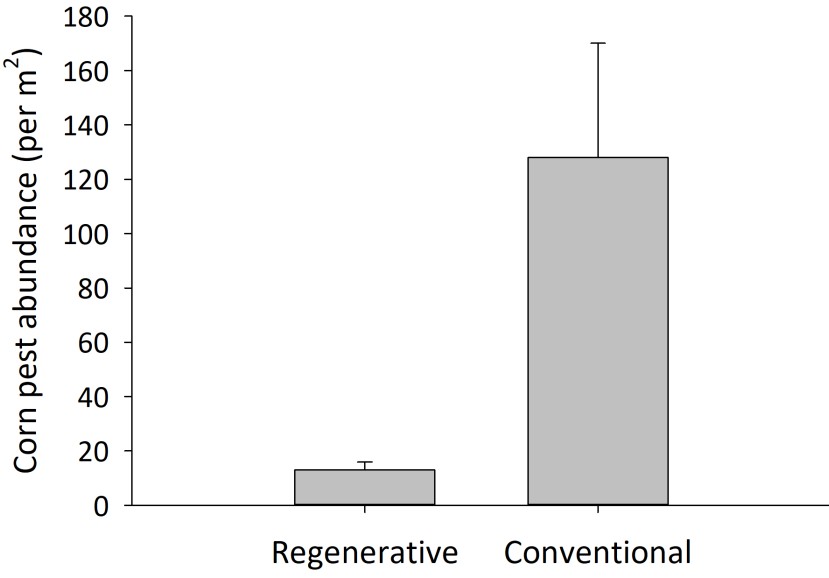

**Figure 1 Insecticide-treated cornfields had higher pest abundance than untreated, regenerative cornfields.** Values presented are mean ± SEM total pests (corn rootworm adults, European corn borers, Western bean cutworm, other caterpillars, and aphids) per m², and were assessed during corn anthesis. The systems were regarded as best-management practices for the sampled region by the farmers themselves. All conventional farms planted neonicotinoid-treated, Bt corn seed to prophylactically reduce pests, and some cornfields were also sprayed with insecticides. Regenerative farms included >3 of the following practices: use of a multispecies cover crop, abandonment of insecticide, abandonment of tillage, and the cropland was grazed, etc. Pest abundance was significantly different in the two systems ($\alpha = 0.05$; $n = 39$ regenerative cornfields and 40 conventional cornfields).

damaging levels, as observed in other work in the region (*Hutchison et al., 2010*; *Lundgren et al., 2015*). Pest problems in agriculture are often the product of low biodiversity and simple community structure on numerous spatial scales (*Tscharntke et al., 2012*). Hundreds of invertebrate species have been inventoried from cornfields of the Northern Plains of the US (*Lundgren et al., 2015*; *Welch & Lundgren, 2016*), but these communities represent only 25% of the insect species that lived in ancestral habitats (e.g., prairie) that cornfields replaced in this region (*Schmid et al., 2015*). Pest abundance is lower in cornfields that have greater insect diversity, enhanced biological network strength and greater community evenness (*Lundgren & Fausti, 2015*). Suggested mechanisms to explain how invertebrate diversity and network interactions reduce pests include predation (*Letourneau et al., 2009*), competition (*Barbosa et al., 2009*), and other processes that may not be easily predicted. What practices foster diversity in agroecosystems? In our studies, farmers that replaced insecticide use with agronomic forms of plant diversity invariably had fewer pest problems than those with strict monocultures. Reducing insect diversity and relying solely on insecticide use establishes a scenario whereby pests persist and resurge through adaptation, as was observed by our forebears (*Perkins, 1982*; *Stern et al., 1959*). Applying winter cover crops (*Lundgren & Fergen, 2011*), lengthening crop rotations (*Bullock, 1992*), diversifying field margins using conservation mixes (*Haaland, Naisbit & Bersier, 2011*), and allowing or promoting non-crop plants between crop rows (*Khan et al., 2006*) are other agronomically

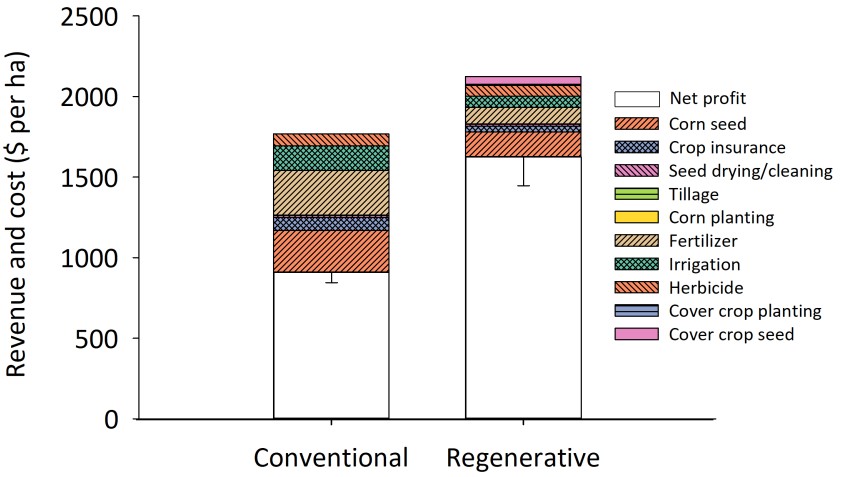

**Figure 2 Regenerative corn fields generate nearly twice the profit of conventionally managed corn fields.** The heights of the bars represent average gross profits across all 40 fields (in each treatment). Profit was calculated using direct costs and revenues for each field and excludes any overhead and indirect expenses. Regenerative cornfields implemented three or more practices such as planting a multispecies cover mix, eliminating pesticide use, abandoning tillage, and integrating livestock onto the crop ground. Conventional cornfields used fewer than two of these practices. The regenerative systems had 70% higher profit than conventional cornfields ($\alpha = 0.05$; $n = 36$ fields in each system). Seed drying, corn planting, and cover crop planting are present on the graphs, but were negligible costs.

sound practices that regenerative farmers successfully apply to improve the resilience of their system to pest proliferation.

Despite having lower grain yields, the regenerative system was nearly twice as profitable as the conventional corn farms (ANOVA; $F_{1,70} = 14.35$, $P < 0.001$; Fig. 2). Regenerative farms produced 29% less corn grain than conventional operations ($8,481 \pm 684$ kg/ha vs. $11,884 \pm 648$ kg/ha; ANOVA; $F_{1,70} = 8.39$, $P = 0.01$). Yield reductions are commonly reported in more ecologically based food production systems relative to conventional systems (*De Ponti, Rijk & Van Ittersum, 2012*). However, only 4% of calories produced as corn grain is eaten directly by humans, and almost none is consumed as grain. Thirty-six percent of grain is fed to livestock (*NASS, 2017*), and corn-fed beef contains only 13% of the total calories produced by corn grain. Two ways that regenerative systems could increase the human food produced per ha in cornfields would be to increase the diversity of livestock on the field, or increasing the duration of grazing current stock. The relative profitability in the two systems was driven by the high seed and fertilizer costs that conventional farms incurred (32% of the gross income went into these inputs on conventional fields, versus only 12% in regenerative fields), and the higher revenue generated from grain and other products produced (e.g., meat production) on the regenerative corn fields (Fig. 2). The high seed costs on conventional farms are largely attributable to premiums paid by farmers for prophylactic insecticide traits (no insecticide was applied as spray on these fields), whose value is questionable due to pest resistance and persistent low abundance for some targeted pests in the Northern Plains (*Hutchison et al., 2007*; *Krupke et al., 2017*). Regenerative farmers reduced their fertilizer costs by including legume-based cover crops
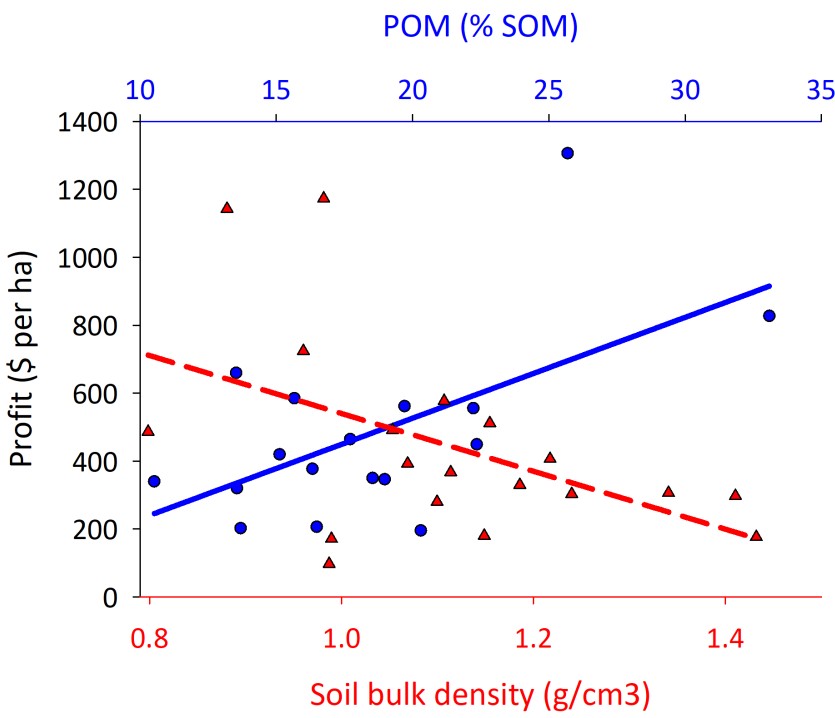

**Figure 3 Corn fields with high particulate organic matter and low bulk density in the soil have greater profits.** Corn fields were managed under either conventional or regenerative systems, and profit was calculated using direct costs and revenues for each field and excludes any overhead and indirect expenses. (general linear regression model; $F_{1,16} = 7.84$; $P = 0.01$; $r^2 = 0.34$; profit $= 29.68$[POM]–66.94; bulk density; $F_{1,19} = 5.23$; $P = 0.03$; $r^2 = 0.24$; profit $= -975$ [POM] $+ 1,593$).

on their fields during the fallow period (*Ebelhar, Frye & Blevins, 1984*), adopting no-till practices (*Lal, Reicosky & Hanson, 2007*), and grazing the crop field with livestock (*Russelle, Entz & Franzluebbers, 2010*). They also received higher value for their crop by receiving an organic premium, by selling their grain directly to consumers as seed or feed, and by extracting more than just corn revenue from their field (e.g., by grazing cover mixes with livestock).

The profitability of a corn field was not related to grain yields ($F_{1,70} < 0.001$; $P = 0.98$; $r^2 < 0.01$; profit $= -0.0006$[yield] $+ 1,274$), but was positively correlated with the level of POM in the soil, and inversely related to the bulk density of the soil (Fig. 3; the SOM quantities upon which %POM are presented here are reported in Table 2). Organic matter is considered by some as the basis for productivity in the soil (*Karlen et al., 1997*; *Tiessen, Cuevas & Chacon, 1994*), and soils with high SOM typically have lower bulk density. SOM increases water infiltration rates, and supports greater microbial and animal abundance and diversity (*Lehman et al., 2015*). The components of POM are the labile portion of this SOM, and are frequently used to study the effects of management-based differences in SOM (*Cambardella & Elliott, 1992*). The only way to generate SOM *in situ* in cropland is through fostering biology, which inherently is driven by plant communities through sequestration of $CO_2$ from the atmosphere. Eliminating tillage (*Pikul Jr et al., 2007*; *Six, Elliott & Paustian,*

**Table 2  Soil organic matter on regenerative and conventional corn farms.** Shaded rows represent regenerative corn farms.

| Reference town | Farm locations (latitude, longitude) | SOM (%) |
| --- | --- | --- |
| Bladen, NE | 40.31971, −98.57358 | 6.23 |
| Bladen, NE | 40.33703, −98.56301 | 4.52 |
| York, NE | 40.63054, −97.66534 | 6.21 |
| York, NE | 40.97390, −97.49031 | 5.55 |
| Bismarck, ND | 46.85280, −100.60131 | 4.19 |
| Bismarck, ND | 46.85280, −100.35145 | N/A |
| Bismarck, ND | 46.81734, −100.51257 | 5.82 |
| Bismarck, ND | 47.14250, −100.19720 | 3.85 |
| White, SD | 44.42572, −96.58806 | N/A |
| White, SD | 44.41155, −96.60008 | 5.52 |
| Pipestone, MN | 44.11446, −96.32468 | N/A |
| Pipestone, MN | 44.12416, −96.36422 | 4.75 |
| Toronto, SD | 44.59248, −96.57923 | 7.60 |
| Toronto, SD | 44.57960, −96.58367 | 6.38 |
| Gary, SD | 44.80565, −96.34708 | 7.53 |
| Gary, SD | 44.80689, −96.35465 | 7.36 |
| Arlington, SD | 44.41566, −97.18795 | 8.17 |
| Arlington, SD | 44.42644, −97.25077 | 8.18 |
| Lake Norden, SD | 44.58976, −97.08649 | 4.56 |
| Lake Norden, SD | 44.55.6839, −97.243820 | 6.26 |

*1999*), implementing cover crops (*Ding et al., 2006*; *Kuo, Sainju & Jellum, 1997*), and cycling plant residue through livestock (*Tracy & Zhang, 2008*) all enhance this process, and all are important practices used in regenerative food systems that raise POM in the soil.

## CONCLUSIONS

The farmers themselves have devised an ecologically based production system comprised of multiple practices that are woven into a profitable farm that promotes ecosystem services. Regenerative farms fundamentally challenge the current food production paradigm that maximizes gross profits at the expense of net gains for the farmer. Key elements of this successful approach to farming include

1. By promoting soil biology and organic matter and biodiversity on their farms, regenerative farmers required fewer costly inputs like insecticides and fertilizers, and managed their pest populations more effectively.
2. Soil organic matter was a more important driver of proximate farm profitability than yields were, in part because the regenerative farms marketed their products differently or had a diversified income stream from a single field.

## ACKNOWLEDGEMENTS

We thank our 20 farmers throughout the Northern Plains for providing us with study sites and management information. E Adee, M Bredeson, J Fergen, D Grosz, K Januschka, N Koens, R LaCanne, M La Vallie, A Leiferman, J Lundgren, A Martens, C Mogren, K Nemec, A Nikolas, J Pecenka, G Schen, C Snyder, & K Weathers assisted field work. R Conser, M Entz, C Morrissey, & R Teague provided comments on earlier drafts. M Longfellow and L Hesler identified invertebrates. Mention of trade names or commercial products in this publication does not imply recommendation or endorsement by South Dakota State University or Ecdysis Foundation.

### Funding

The project was supported by USDA PMAP Award # 2013-34381-21245, a NC-SARE graduate student fellowship GNC16-227, and donations of farmers and beekeepers to Ecdysis Foundation. The funders had no role in study design, data collection and analysis, decision to publish, or preparation of the manuscript.

### Grant Disclosures

The following grant information was disclosed by the authors:
USDA PMAP Award: #2013-34381-21245.
NC-SARE: GNC16-227.
Ecdysis Foundation.

### Competing Interests

Jonathan G. Lundgren is the CEO for Blue Dasher Farm and director of the Ecdysis Foundation. Claire E. LaCanne is an employee of the University of Minnesota, and was a graduate student for South Dakota State University during her thesis program (this work is part of that thesis).

### Author Contributions

- Claire E. LaCanne conceived and designed the experiments, performed the experiments, analyzed the data, prepared figures and/or tables, authored or reviewed drafts of the paper, approved the final draft.
- Jonathan G. Lundgren conceived and designed the experiments, analyzed the data, contributed reagents/materials/analysis tools, prepared figures and/or tables, authored or reviewed drafts of the paper, approved the final draft.

### Data Availability

  The raw data is provided as a Supplemental File.

## Supplemental Information

Supplemental information for this article can be found online at http://dx.doi.org/10.7717/peerj.4428#supplemental-information.

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
