# Peer review of "Regenerative agriculture: merging farming and natural resource conservation profitably"

_PeerJ, doi:10.7717/peerj.4428_

## Round 0.1 · original submission · Minor Revisions

Please see the comments below from the two reviewers. Please incorporate them or address them. I look forward to receiving the revised manuscript.

Reviewer 1 ·

Basic reporting

This manuscript is clearly written with simple appropriate language. The references are adequate throughout and the figures, raw data and structure are all reasonable.

I would not that in figure 2 there is no description for those data represented by the white bars

Experimental design

This research is field based and down from various sources on a farm level. Given the nature of the data used, there is an inherent limit to what how much actual experimental design and manipulation is possible. Within those confines, the authors did a commendable job of gathering their data. The methods for collecting insects etc. are quite sound and the ability to locate 20 farms and gain the relevant information from each makes for quite reasonable sample sizes.

Validity of the findings

The findings of this study are quite sound and reasonable given the data and analyses. The conclusions are sound and within the scope of those data collected. There is some reasonable speculation, and while this study is limited in its ability to provide mechanism, the speculation is aimed at these mechanisms which is appropriate.

·

Basic reporting

Acceptable

Experimental design

The paired design is implied in Table S1 but not clear in the Methods. Please clarify statistical design. See comments on the manuscript (attached).

L 74 - how were fields selected on each farm? Why were 2 omitted?
L 81 - how were the sites elected for yield determination?
L 83 - how was revenue for livestock grazing determined?
L 86 - Presumably numbers of indiv from all spp are mentioned. Clarify.
How many of each sp? Did 1 sp do more damage than another? Was crop damage due to pests assessed?
L 116-17 - how was the value of other products assessed?
L 118 - Extra cost only for seed or also for direct applications of insecticide?
L 123 - how many regen farmers received an organic premium? Show in T S1

Validity of the findings

T S2 - more explanation needed in order for table to stand alone. Stats related to T S2? Not referred to in text
Fig 2 - see comment bubbles on Fig 2. It's not clear to me what this shows. It appears that some costs are missing.

Additional comments

This type of systems analysis is required and appreciated.
Please address comments above in order to help readers understand how data were collected and analyzed.

---

## Round 0.2 · accepted · Accept

Thank you for incorporating the suggested revisions.